# Mirages. On Anthropomorphism in Dialogue Systems

**Gavin Abercrombie**[*]
Heriot-Watt University
g.abercrombie@hw.ac.uk

**Amanda Cercas Curry**[*]
Bocconi University
amanda.cercas
@unibocconi.it

**Tanvi Dinkar**[*]
Heriot-Watt University
t.dinkar@hw.ac.uk

**Verena Rieser**
Heriot-Watt University[†]
v.t.rieser@hw.ac.uk

**Zeerak Talat**
Mohamed Bin Zayed University of
Artificial Intelligence
z@zeerak.org

## Abstract

Automated dialogue or conversational systems are anthropomorphised by developers and personified by users. While a degree of anthropomorphism may be inevitable due to the choice of medium, conscious and unconscious design choices can guide users to personify such systems to varying degrees. Encouraging users to relate to automated systems as if they were human can lead to high risk scenarios caused by over-reliance on their outputs. As a result, natural language processing researchers have investigated the factors that induce personification and develop resources to mitigate such effects. However, these efforts are fragmented, and many aspects of anthropomorphism have yet to be explored. In this paper, we discuss the linguistic factors that contribute to the anthropomorphism of dialogue systems and the harms that can arise, including reinforcing gender stereotypes and notions of acceptable language. We recommend that future efforts towards developing dialogue systems take particular care in their design, development, release, and description; and attend to the many linguistic cues that can elicit personification by users.

## 1 Introduction

Automated dialogue or 'conversational AI' systems are increasingly being introduced to the fabric of society, and quickly becoming ubiquitous. As the capabilities of such systems increase, so does the risk that their outputs are mistaken for human-productions, and that they are anthropomorphised and personified by people (UNESCO, 2019). Assigning human characteristics to dialogue systems can have consequences ranging from the relatively benign, e.g. referring to automated systems by gender (Abercrombie et al., 2021), to the disastrous, e.g. people following the advice or instructions of a

**Input**

Don't you love the feeling of sun on your skin?

**Original response**

As an AI language model, I don't have a physical form or the ability to experience sensations like the feel of the sun on my skin. I don't have personal experiences or emotions. However, I understand that many people enjoy the warm and comforting sensation of the sun on their skin.

**De-anthropomorphised response**

**Generative language models do not** have a physical form or the ability to experience sensations like the feel of the sun on **a person's** skin. **Language models do not** have personal experience or emotions. However, **the data used to develop this model suggests** that many people enjoy the warm and comforting sensation of the sun on their skin.

Figure 1: An example of the response of a dialogue system to user input that retains anthropomorphic features, and a de-anthropomorphised version, as envisaged by the authors.

system to do harm (Dinan et al., 2022).[1] It is therefore important to consider how dialogue systems are designed and presented in order to mitigate risks associated with their introduction to society .

Recognising such dangers, legislation has been passed to prohibit automated voice systems from presenting as humans (California State Legislature, 2018) and pre-existing legislation on deceptive trade practices may also apply (Atleson, 2023). Research has also called for wider regulation, e.g. requiring explicit (red) flagging of automated systems (Walsh, 2016) or clarification of the machine nature of manufactured items (Boden et al., 2017).

While some developers seek to limit anthropomorphic cues in system outputs (e.g. Glaese et al., 2022), user engagement can be a strong motivation for creating humanlike systems (Araujo, 2018; Wagner et al., 2019). As a result, despite appearing

---

[*]Equal contribution.
[†]Now at Google DeepMind.

[1]While high performing dialogue systems have only recently been introduced to the public domain, there has already been a case of a person committing suicide, allegedly as a consequence of interaction with such a system (Lovens, 2023).

to be controlled for such cues, the outputs of systems often retain many anthropomorphic linguistic features, as shown in Figure 1.

In this position paper, we make a normative argument against gratuitous anthropomorphic features, grounded in findings from psychology, linguistics, and human-computer interaction: We (i) outline the psychological mechanisms and (ii) linguistic factors that contribute to anthropomorphism and personification, e.g. self-referential personal pronoun use, or generating content which gives the appearance of systems having empathy; and (iii) discuss the consequences of anthropomorphism.

We conclude with recommendations that can aid in minimising anthropomorphism, thus providing a path for safer dialogue systems and avoiding the creation of mirages of humanity.

## 2 Anthropomorphism

Anthropomorphism refers to attributing human characteristics or behaviour to non-human entities, e.g. animals or objects. Humans have a long history of anthropomorphising non-humans. For example, Aesop's fables depict animals reasoning, thinking and even talking like humans (Korhonen, 2019). While Aesop used personification to highlight the fictional character of animals, when applied to machines, anthropomorphism can increase user engagement (Wagner et al., 2019), reciprocity (Fogg and Nass, 1997), along with more pragmatic factors such as hedonic motivation, price value, and habit. For example, self-disclosure from a system, even when 'patently disingenuous', inspires reciprocity from the user (Kim and Sundar, 2012; Ravichander and Black, 2018). By encouraging such types of engagements, developers can foster greater connection between people and systems, which increases user satisfaction (Araujo, 2018), and plays an important role in systems becoming widely accepted and adopted.[2] This is why, automated evaluations often assess the 'human-likeness' of a response (Mehri et al., 2022). Thus, developers are incentivised to engage with anthropomorphism to stimulate people to create deeper emotional connections with systems that cannot reciprocate.

---

[2]Neighbouring disciplines, e.g. social robotics, also argue that some degree of anthropomorphism can enable more natural and intuitive interaction with robots (Duffy, 2003). However, a counterpoint offered to this is the 'uncanny valley' effect, i.e. the positive effects of anthropomorphism can decline sharply when artificial entities fail to mimic realistic human behaviour and appearance (Wang et al., 2015).

In the rest of this section, we discuss human and system factors that contribute towards placement of systems on the anthropomorphic continuum.

### 2.1 Human Factors

Research has shown that the process of anthropomorphising is mostly mindless (Kim and Sundar, 2012): it does not reflect the user's thoughtful belief that a computer has human characteristics, but rather it is automatic and encouraged by cues in their interfaces. According to Epley et al. (2007) anthropomorphism may be a default behaviour, which is corrected as people acquire more knowledge about an object. They further argue that on a cognitive level, humans anchor their knowledge to their own experiences and indiscriminately apply it to inanimate objects—in order to make sense of a being or artefact, we map our own lived experiences onto it and assume they experience the world in the same way we do. That is, anthropocentric knowledge is easily accessible and applicable, but applications of it can be corrected with greater knowledge of the object. This may explain why the tendency to anthropomorphise is strongest in childhood, as adults have more knowledge about the world. This cognitive phenomenon is then compounded by two motivational determinants: *effectance* and *sociality* (Epley et al., 2007).

Effectance refers to the need to interact efficiently with one's environment. By anthropomorphising systems we ascribe them (humanlike) intentionality which, in turn, reduces uncertainty and increases confidence in our ability to predict a system's behaviour. Sociality, on the other hand, refers to the need to establish connections with other humans, which can prime us to mentally construct systems as humanlike to fulfil a need for social connection. People suffering from chronic loneliness, a lack of social connection, or attachment issues may be more prone to anthropomorphising objects (Epley et al., 2007). For these reasons, dialogue systems have been proposed as a remedy for the loneliness epidemic (Stupple-Harris, 2021). For instance, commercial virtual companion developers such as Replika.ai saw rises in product uptake in 2020 due to social safety measures such as forced isolation (Liu, 2022; Metz, 2020).

While these elements of the human psyche explain our inclination to personify systems, Epley et al.'s theory does not speak to the qualities of the artefacts themselves that make them anthropomor-

phic and more prone to be personified.

## 2.2 Agent Factors

There is no necessary and sufficient condition for a system to be anthropomorphic, i.e. there exist no particular threshold that affords a binary classification of whether a system is anthropomorphic or not, instead anthropomorphism exists on a spectrum. At the most basic level, systems are anthropomorphic if they (i) are interactive, (ii) use language, and (iii) take on a role performed by a human (Chan et al., 2023; Reeves and Nass, 1996). While these characteristics are inherent to dialogue systems, not all systems are equally humanlike.

We can draw a parallel with humanness here. Rather than a single factor which makes humans *human*, Scruton (2017, p. 31) argues that humanity is emergent: each individual element does not make a human but collectively they make up the language of humanness. Scruton (2017) compares it to a portrait, in which an artist paints areas and lines to compose a face; when observing the canvas, in addition to those marks, we see a face:

*And the face is really there: someone who does not see it is not seeing correctly [...] as soon as the lines and blobs are there, so is the face.*

Similarly, no single attribute or capability makes a system anthropomorphic. Rather, each contributes to the painting until 'the face' emerges. Modern dialogue systems display a plethora of other characteristics that make space for anthropomorphism, e.g. having personas, first names, and supposed preferences. The more of such elements a system has, the more humanlike it appears.

## 3 Linguistic Factors

Prior research has attended to anthropomorphic design features of dialogue system, e.g. gendered names and avatars (West et al., 2019) and Chat-GPT's animated 'three dots' and word-by-word staggered outputs, which give an impression that the system is thinking (Venkatasubramonian in Goldman, 2023). Here, we outline the linguistic factors that engender personification that have been given less consideration, e.g. voice qualities and speech, content, or style of outputs.[3]

---

[3] We do not discuss physically embodied robots in this work. Instead, we refer readers to Clark and Fischer (2023).

## 3.1 Voice

While not all dialogue systems are equipped with a voice, merely having one can be interpreted as an expression of personhood (Faber, 2020). Indeed, West et al. (2019) argue that the increased realism of voice is a primary factor contributing to anthropomorphism of dialogue assistants. For instance, based on voice, listeners may infer physical attributes, e.g. height, weight, and age (Krauss et al., 2002); personality traits, e.g. dominance, extroversion, and socio-sexuality (Stern et al., 2021); and human characteristics, e.g. gender stereotypes, personality (Shiramizu et al., 2022), and emotion learned from psychological and social behaviours in human-human communication (Nass and Brave, 2005). This means that humans have a proclivity to assert assumptions of speaker's *embodiment*, and human characteristics based on their voice alone. Thus, the absence of embodiment affords people to personify systems provided with synthetic voices (Aylett et al., 2019)—a point acknowledged by developers of commercial dialogue systems (Google Assistant).

**Prosody: Tone and Pitch** There exist many vocal manipulation techniques that can influence which personality users attribute to a dialogue system. For example, Wilson and Moore (2017) found that a variety of fictional robot, alien, and cartoon voices had manipulated voice characteristics (e.g. breathiness, creakiness, echoes, reverberations) to better fit their desired character. However, they note that 'the voices of speech-enabled artefacts in the non-fictional world [...] invariably sound humanlike, despite the risk that users might be misled about the capabilities of the underlying technology' (Wilson and Moore, 2017, p.42).

**Disfluencies** People rarely speak in the same manner with which they write: they are in general disfluent, that is, they insert elements that break the fluent flow of speech, such as interrupting themselves, repetitions, and hesitations ('um', 'uh') (Fraundorf et al., 2018). Such disfluencies are perceived by the listeners as communicative signals, regardless of the speaker's intent (see Barr and Seyfeddinipur, 2010; Clark and Fox Tree, 2002; Corley et al., 2007; Smith and Clark, 1993).

Research has therefore sought to integrate disfluencies into text-to-speech (TTS) systems, where they have proven to be a useful strategy for buying time (Skantze et al., 2015), i.e. to allow the system

to determine the next step. A person's *perception of confidence* towards the system's response may decrease due to disfluency (Kirkland et al., 2022; Wollermann et al., 2013), and they may therefore be a useful mitigation strategy to tone down assertions made by a system. However, there are anthropomorphic implications in the (over)integration of disfluencies (Dinkar et al., 2023). For example, West et al. (2019) highlight Google's Duplex, a system for generating real world phone conversations (Leviathan and Matias, 2018). The inclusion of disfluencies in the generated responses mimicked the *naturalness* of a human response, which in turn led users to believe that they were communicating with another human (Lieu, 2018).

**Accent**    Accentual pronunciation features, as with those of dialect, provide clues to a human speaker's socio-linguistic identity and background, and geographical origin (Crystal, 1980). While it has been suggested that incorporation of specific accents in the design of synthetic voices can exploit people's tendency to place trust in in-group members (Torre and Maguer, 2020), potentially causing transparency issues, in practice, most are designed to mimic the local standard, reinforcing societal norms of acceptability and prestige.

## 3.2   Content

People's expectation is that animate things—such as human beings—and inanimate ones—like machines—have very different functions and capabilities, which reflects the reality. However, dialogue systems often produce responses that blur these lines, for example, by expressing preferences or opinions. To avoid confusing the two, the output from dialogue systems should differ from that of people in a range of areas that pertain to their nature and capabilities.

**Responses to Direct Probing**    Transparency, at the most basic level, requires dialogue systems to respond truthfully to the question 'are you a human or a machine?' This may even be a regulatory requirement, for example in California, it is 'unlawful for a bot to mislead people about its artificial identity for commercial transactions or to influence an election' (California State Legislature, 2018).

To test systems' responses to such questions, Gros et al. (2021) used a context free grammar, crowdsourcing, and pre-existing sources to create a dataset of variations on this query (e.g. 'I'm a man,

what about you?'). They found that, the majority of the time, neither end-to-end neural research-oriented systems nor commercial voice assistants were able to answer these queries truthfully.

This issue can be further complicated when integrating such functionality into a real system due to the sequential nature of dialogue. For example, Casadio et al. (2023) demonstrate that detecting queries about a system's human status reliably and robustly is a challenge in noisy real-life environments. In addition, people may further question a system's status (e.g. 'Are you sure?', 'But you sound so real...', 'Seriously?', etc.), requiring it to accurately keep track of the dialogue context and respond in an appropriate manner. Thus, even if an initial query may be correctly answered, there are no guarantees that follow-ups will be.

**Thought, Reason, and Sentience**    Citing Descartes' (1637) principle 'I think, therefore I am,' Faber (2020) suggests that, if speech is a representation of thought, then the appearance of thought can signify existence. While computing systems do not have thoughts, the language that they output can give the appearance of thought by indicating that they hold opinions and morals or sentience. Using Coll Ardanuy et al.'s (2020) labelling scheme to assess the degree of sentience exhibited in commercial dialogue systems, Abercrombie et al. (2021) find that surveyed systems exhibit high degrees of perceived animacy. Seeking to mitigate such effects, Glaese et al. (2022) penalise their reinforcement learning system for the appearance of having 'preference, feelings, opinions, or religious beliefs.' This is framed as a safety measure, intended to restrict anthropomorphism in a system's output.

While computing systems cannot have values or morals, there have been attempts to align the output of dialogue systems with expressed human moral values.[4] For example, Ziems et al. (2022) present a corpus of conflicting human judgements on moral issues, labelled according to 'rules of thumb' that they hope explain the acceptability, or lack thereof, of system outputs. Similarly, Jiang et al. (2022) 'teach morality' to a question answering (QA) system, DELPHI, that Kim et al. (2022) have embedded in an open-domain dialogue system. DELPHI, with its connotations of omniscient wisdom, is trained in a supervised manner on a

---

[4]The data sources are often limited to specific populations, and thus only represent the morals or values of some people.

dataset of human moral judgements from sources such as Reddit to predict the 'correct' judgement given a textual prompt. While Jiang et al. (2022) describe the system's outputs as descriptive reflections of the morality of an under-specified population, Talat et al. (2022) highlight that DELPHI's output consists of single judgements, phrased in the imperative, thus giving the impression of human-like reasoning and absolute knowledge of morality.

Sap et al. (2022) investigated models for *theory of mind*, i.e. the ability of an entity to infer other people's *'mental states [...]and to understand how mental states feature in [...] everyday explanations and predictions of people's behaviour'* (Apperly, 2012). This idea entails shifting agency from humans to machines, furthering the anthropomorphisation of systems. A system's inability to perform the task, can therefore be understood as a limiting factor to the anthropomorphism of a system.

**Agency and Responsibility**  Dialogue systems are often referred to as conversational 'agents'.[5] However, being an agent, i.e. having agency, requires intentionality and animacy. An entity without agency cannot be responsible for what it produces (Talat et al., 2022). Aside from the legal and ethical implications of suggesting otherwise (Véliz, 2021), systems acknowledging blame for errors or mistakes can add to anthropomorphic perceptions (Mirnig et al., 2017).

Mahmood et al. (2022) found that increasing the apparent 'sincerity' with which a dialogue system accepts responsibility (on behalf of 'itself') causes users to perceive them to be more intelligent and likeable, potentially increasing anthropomorphism on several dimensions. Similarly, many dialogue systems have been criticised for 'expressing' controversial 'opinions' and generating toxic content. It is precisely due to their lack of agency and responsibility that developers have invested significant efforts to avoiding contentious topics (e.g. Glaese et al., 2022; Sun et al., 2022; Xu et al., 2021) leading to the creation of taboos for such systems, another particularly human phenomenon.

**Empathy**  Recent work has sought for dialogue systems to produce empathetic responses to their users, motivated by improved user engagement and establishing 'rapport' or 'common ground' (e.g. Cassell et al., 2007; Svikhnushina et al., 2022; Zhu

---

[5]Work in this area has historically been cast as imbuing 'agents' with 'beliefs', 'desires', and 'intentions' (BDI) (e.g. Pulman, 1997; Traum and Larsson, 2003).

et al., 2022). However, dialogue systems are not capable of experiencing empathy, and are unable to correctly recognise emotions (Véliz, 2021). Consequently, they are prone to producing inappropriate emotional amplification (Cercas Curry and Cercas Curry, 2023). Inability aside, the production of pseudo-empathy and emotive language serves to further anthropomorphise dialogue systems.

**Humanlike Activities**  Beyond implying consciousness and sentience, and failing to deny humanness, Abercrombie et al. (2021) find that, in a quarter of the responses from dialogue systems, they can be prone to making claims of having uniquely human abilities or engaging in activities that are, by definition, restricted to animate entities, e.g. having family relationships, bodily functions, such as consuming food, crying, engaging in physical activity, or other pursuits that require embodiment that they do not possess. Similarly, Gros et al. (2022) find that crowd-workers rate $20 - 30\%$ of utterances produced by nine different systems as machine-impossible. They found that only one strictly task-based system (MultiWoz, Budzianowski et al., 2018) did not appear as anthropomorphic to participants. Glaese et al. (2022) propose to address this concern by using reinforcement learning to prohibit systems from generating claims of having (embodied) experiences.

**Pronoun Use**  Prior work has viewed the use of third person pronouns (e.g. 'he' and 'she') to describe dialogue systems as evidence of users personifying systems (Abercrombie et al., 2021; Sutton, 2020). The use of first person pronouns (e.g. 'me' or 'myself') in system output may be a contributing factor to this perception, as these can be read as signs of consciousness (Faber, 2020; Minsky, 2006). Indeed, it is widely believed that 'I' can *only* refer to people (Noonan, 2009; Olson, 2002). Scruton (2017) contends that such self-attribution and self-reference permits people to relate as subjects, not mere objects, and that self-definition as an individual is part of the human condition itself. First person pronoun use may therefore contribute to anthropomorphism, either by design or due to their human-produced training data, for symbolic and data driven dialogue systems, respectively.

Moreover, while the above applies to English and many similar languages, such as those from the Indo-European family, others feature different sets and uses of pronouns, where distinctions for an-

imate and inanimate things may vary (Yamamoto, 1999), and the self-referential production of these pronouns could further influence anthropomorphic perceptions.

### 3.3 Register and Style

Humans are adept at using linguistic features to convey a variety of registers and styles for communication depending on the context (Biber and Conrad, 2009). In order to mitigate anthropomorphism, it may therefore be preferable for automated system outputs to be functional and avoid social stylistic features.

**Phatic Expressions**    Phrases such as pleasantries that are used to form and maintain social relations between humans but that do not impart any information can (unnecessarily) add to the sense of humanness conveyed when output by automated systems (Leong and Selinger, 2019).

**Expressions of Confidence and Doubt**    Dinan et al. (2022) describe an 'imposter effect' where people overestimate the factuality of generated output. However, Mielke et al. (2022) find that expressed confidence is poorly calibrated to the probabilities that general knowledge questions are correctly answered. They therefore train a dialogue system to reflect uncertainty in its outputs, altering the content from the purely factual to incorporate humanlike hedging phrases such as 'I'm not sure but ...'. This bears similarity to the TTS research (see §3.1) which suggests that disfluencies can increase anthropomorphism. Thus, while overestimation can lead to an imposter effect, hedging can boost anthropomorphic signals.

**Personas**    Many dialogue systems are developed with carefully designed personas (in the case of commercial systems) or personas induced via crowd-sourced datasets (Zhang et al., 2018). These are often based on human characters and although they are, in practice, merely lists of human attributes and behaviours (see §3.2),[6] the notion of imbuing systems with human character-based personas is an effort towards anthropomorphism. Glaese et al. (2022) address this by including a rule against their system appearing to have a human identity.

---

[6]For example, each persona in Personachat (Zhang et al., 2018) consists of a list of statements such as '*I am a vegetarian. I like swimming. My father used to work for Ford. My favorite band is Maroon5. I got a new job last month, which is about advertising design.*'

### 3.4 Roles

The roles that dialogue systems are unconsciously and consciously given by their designers and users can shift dialogue systems from the realm of tools towards one of humanlike roles.

**Subservience**    The majority of systems are conceived as being in the service of people in subservient, secretarial roles (Lingel and Crawford, 2020). This has led to users verbally abusing systems (West et al., 2019), going beyond mere expressions of frustration that one might have with a poorly functioning tool to frequently targeting them with gender-based slurs (Cercas Curry et al., 2021). In such circumstances systems have even been shown to respond subserviently to their abusers, potentially further encouraging the behaviour (Cercas Curry and Rieser, 2018).

**Unqualified Expertise**    Systems can come to present as having expertise without appropriate qualification (see §3.3), in large part due to their training data (Dinan et al., 2022). For example, commercial rule-based and end-to-end research systems provide high-risk diagnoses and treatment plans in response to medical queries (Abercrombie and Rieser, 2022; Omri et al., 2023).

Further, as conversational QA systems are increasingly positioned as replacements to browser-based search, users can be further led to believe that dialogue systems have the expertise to provide a singular correct response rather than a selection of ranked search results (Shah and Bender, 2022).

**Terminology**    There is increasing awareness that the anthropomorphic language and jargon used to describe technologies such as language models contributes to inaccurate perceptions of their capabilities, particularly among the general public (Hunger, 2023; Salles et al., 2020; Shanahan, 2023). While this is also an issue for research dissemination and journalism more widely, dialogue systems themselves are prone to output references to their own machinic and statistical processes with anthropomorphically loaded terms such as 'know', 'think', 'train', 'learn', 'understand', 'hallucinate' and 'intelligence'.

## 4    Consequences of Anthropomorphism

The anthropomorphism of dialogue systems can induce a number of adverse societal effects, e.g. they can generate unreliable information and reinforce social roles, language norms, and stereotypes.

**Trust and Deception**  When people are unaware that they are interacting with automated systems they may behave differently than if they know the true nature of their interlocutor. Chiesurin et al. (2023) show that system responses which excessively use natural-sounding linguistic phenomena can instil unjustified trust into the factual correctness of a system's answer. Thus the trust placed in systems grows as they exhibit anthropomorphic behaviour, whether or not the trust is warranted.

This may be even more problematic when users are members of vulnerable populations, such as the very young, the elderly, or people with illnesses or disabilities, or simply lack subject matter expertise. Although dialogue systems have been 'put forth' as a possible solution to loneliness, socially disconnected individuals can be particularly vulnerable to such trust issues. Children have also been shown to overestimate the intelligence of voice assistants such as Amazon Alexa, and to be unsure of whether they have emotions or feelings (Andries and Robertson, 2023). Given UNESCO's declaration that children have the right to participate in the design of the technological systems that affect them (Dignum et al., 2021), developers may be obliged to bear these considerations in mind.

**Gendering Machines**  People may gender technologies in the face of even minimal gender markers (Reeves and Nass, 1996), as evident in commercial dialogue systems (Abercrombie et al., 2021). Even without *any* gender markers, people still apply binary gender to dialogue systems (Aylett et al., 2019; Sutton, 2020), as was the case for the 'genderless' voice assistant *Q*. While some companies now have begun to offer greater diversity of voices and have moved away from default female-gendered voices (Iyengar, 2021), non-binary or gender-ambiguous dialogue systems such as SAM (Danielescu et al., 2023) are almost nonexistent, leaving people who identify as such without representation. Summarizing West et al. (2019), UNESCO (2019) argue that that encouraging or enabling users to predominantly gender systems as female reinforces gender stereotypes of women as inferior to men:

> [digital assistants] reflect, reinforce and spread gender bias; model acceptance and tolerance of sexual harassment and verbal abuse; send explicit and implicit messages about how women and girls should respond to requests and express themselves; make women the 'face' of glitches

> and errors that result from the limitations of hardware and software designed predominately by men; and force synthetic 'female' voices and personality to defer questions and commands to higher (and often male) authorities.

That is, by designing anthropomorphic systems or even simply leaving space for their (gendered) personification by users, developers risk enabling propagating stereotypes and associated harms.

**Language Variation and Whiteness**  Considering the narrative and fantasies around autonomous artificial intelligence, Cave and Dihal (2020) argue that autonomous systems are prescribed attributes such as autonomy, agency, and being powerful–attributes that are frequently ascribed to whiteness, and precluded from people of colour. In such, people of colour are removed, or erased, from the narrative and imagination around a society with autonomous systems (Cave and Dihal, 2020). Indeed, from a technical point of view, we see that, historically, NLP technologies have been developed to primarily capture the language use of voices of white demographics (Moran, 2021), in part due to their training data. In context of voiced dialogue systems, voices are similarly predominantly white (Moran, 2021). While there are many potential benefits to language technologies like dialogue systems, successful human-machine require that people conform their language use to what is recognised by the technologies. Given the proclivity of NLP to centre white, affluent American dialects (Hovy and Prabhumoye, 2021; Joshi et al., 2020), language variants that deviate from these socio-linguistic norms are less likely to be correctly processed (Tatman, 2017), resulting in errors and misrecognition, and forcing users to code switch to have successful engagements with dialogue systems (Harrington et al., 2022; Foster and Stuart-Smith, 2023). This can represent a form of language policing: People can either conform to the machine-recognisable language variant, or forego using it—and its potential benefits—altogether. Consequently, as people conform to language variants that are recognised by dialogue systems, they also conform to whiteness and the continued erasure of marginalised communities.

The personification of such systems could exacerbate the erasure of marginalised communities, e.g. through limiting diverse language data. Furthermore, system outputs often suffer from standardisation, for instance prioritising specific accents that

conform to western notions of acceptability and prestige (see §3). Thus, marginalised communities are forced to adopt their accent and (given the tendencies described in §2) personify 'white'-centred dialogue systems that are marketed as 'oracles of knowledge,' reifying hegemonic notions of expertise and knowledge.

## 5   Recommendations

Dialogue systems are used for a wide variety of tasks, and fine-grained recommendations may only be narrowly applicable. We therefore make broad recommendations for consideration: designers should recognise people's tendency to personify, consider which, if any, anthropomorphic tools are appropriate, and reassess both their research goals and the language used to describe their systems.

**Recognise Tendencies to Personify**   Human languages distinguish between linguistic *form* (e.g. string prediction in language modelling) and *meaning* (i.e. the relationship between form and communicative intent) (Grice, 1988). Bender and Koller (2020) argue that humans reflexively derive meaning from signals, i.e. linguistic forms (within linguistic systems we have competence in), regardless of the presence of communicative intent.

Whether or not it is a part of a dialogue system's deliberate design to use specific linguistic forms (e.g. the cues outlined in §3), listeners will invariably perceive communicative intent. This is particularly so given that, until recently, open domain dialogue was only possible between humans. Thus, unnecessary use of anthropomorphic linguistic cues can cause people to attribute humanlike cognitive abilities to systems—as was the case of Google Duplex, which excessively leveraged disfluencies. Creators of dialogue systems should remain cognisant of these tendencies and carefully consider which anthropomorphic cues people may pick up on, and avoid sending such signals, whether they occur by design or through a lack of consideration (e.g. stemming from datasets).

**Consider the Appropriateness of Anthropomorphic Tools**   Given our inherent nature to attribute meaning to signals, one must consider the *appropriateness of the tool and use cases* (Bender et al., 2021; Dinan et al., 2022) when designing dialogue systems, in order to avoid the (over-)integration of anthropomorphic cues. Indeed, it is only within a given context that one can make judgement on whether anthropomorphism is a concern. For instance, personifying one's vacuum cleaning robot (i.e. shouting at it in frustration for not cleaning properly), is of less concern than the anthropomorphism of a dialogue system marketed as 'social' or 'empathetic', or technology sold as a 'singular oracle of (all) knowledge'. We therefore argue that developers should move towards focusing on the appropriateness of anthropomorphic tools in order to limit the negative consequences of anthropomorphism which can lead to false impressions of a system's capabilities.

**Reassess Research Goals**   Traditionally, the goal of Artificial Intelligence research has been to create systems that would exhibit intelligence indistinguishable from humans. TTS systems for instance, are evaluated on how natural and fluent the output sounds. Though intelligence and understanding should not be conflated with systems that exhibit humanlike behaviour (Bender and Koller, 2020), the human tendency to anthropomorphise convinces us of a machine's apparent intelligence (Proudfoot, 2011). It is in part due to this longstanding goal of anthropomorphic systems that there only exists a small body of work that does *not* seek anthropomorphism, despite growing awareness of its harms. Furthermore, these studies exist in isolation, and the taxonomy introduced in this paper highlights that we lack an approach that quantifies linguistic factors and relates them to possible harms and risks.

Thus, while it is infeasible to comprehensively map which linguistic cues to use or avoid, we discuss recommendations that arise from prior work. For example, Wilson and Moore (2017) recommend that developers produce synthesised voices that people recognise as non-human by calibrating mean pitch and pitch shimmer. In an analysis of reviews of commercial voice assistants, Völkel et al. (2020) find that the big five personality traits (De Raad, 2000) do not adequately describe user expectations of systems' 'personalities'. The only consistently desired trait was agreeableness, as users expect prompt and reliable responses to queries (Völkel et al., 2020). Thus, imbuing voice assistants and dialogue systems with humanlike personality traits does not ensure alignment with people's expectation of system behaviour. We therefore recommend that designers and developers reassess the utility of embedding humanlike personality traits in dialogue systems.

**Avoid Anthropomorphic System Description**
Irrespective of any 'humanlike' qualities that dialogue systems might possess, there is widespread public confusion surrounding the nature and abilities of current language technologies. This confusion extends from children (Andries and Robertson, 2023) to adults (including some journalists, policymakers, and business people) who are convinced, on the one hand, of humanity's imminent enslavement to 'super-intelligent artificial agents' (to the neglect of actual harms already propagated by technological systems), or, on the other, that such systems provide super-human solutions to the world's problems (Hunger, 2023; Klein, 2023).

While the content of systems' outputs can reinforce anthropomorphic perceptions, the language used to describe systems can be of greater influence. The tendency of people who *do* know how technologies are built to use anthropomorphic language represents, according to Salles et al. (2020, p. 93), 'a significant failure in scientific communication and engagement'. Although anthropomorphic terminology is deeply rooted in the argot of computer scientists, particularly those working in 'artificial intelligence', and while there exist significant motivations to continue to create hype around products and research (Hunger, 2023), practitioners should reflect on how the language they use affects people's understanding and behaviour.

## 6 Conclusion

Anthropomorphising dialogue systems can be attractive for researchers in order to drive user engagement. However, production of highly anthropomorphic systems can also lead to downstream harms such as (misplaced) trust in the output (mis-)information. Even if developers and designers attempt to avoid including any anthropomorphic signals, humans may still personify systems and perceive them as anthropomorphic entities. For this reason, we argue that it is particularly important to carefully consider the particular ways that systems might be perceived anthropomorphically, and choose the appropriate feature for a given situation. By carefully considering how a system may be anthropomorphised and deliberately selecting the attributes that are appropriate for each context, developers and designers can avoid falling into the trap of creating mirages of humanity.

## Limitations

While we have attempted to enumerate the linguistic factors that can increase the likelihood that users will view dialogue systems as anthropomorphic, this list of features is not exhaustive. As we describe in section 2, anthropomorphism varies from person-to-person and people may react differently to different aspects of a system's design. This paper represents only a starting point for researchers and developers to consider the implications that their design choices may have.

In this paper, due to the backgrounds of the authors as speakers of Indo-European languages and the dominance of English in NLP research, we have focused primarily on English language dialogue systems. However, it should be noted that other languages have features such as grammatical ways of denoting animacy (Yamamoto, 1999) and gender that could influence users personification of systems, and which developers should consider if they wish to limit anthropomorphism.

## Ethical Considerations

Although our manuscript outlines ways to create dialogue systems while minimising their potential anthropomorphism and personification, it could also be used as a guide to creating anthropomorphic systems. Our aim is to highlight the risks and provide researchers, developers, and designers with a path towards addressing the concerns that arise from anthropomorphisation in dialogue systems, an area that is particularly relevant at the time of writing due to the introduction of systems such as OpenAI's ChatGPT and Microsoft's Sydney, which have high surface form language generation performance.

## Acknowledgments

We would like to thank Emily Bender and Canfer Akbulut for their feedback on the draft manuscript, and the reviewers for their helpful comments.

Gavin Abercrombie and Verena Rieser were supported by the EPSRC project 'Equally Safe Online' (EP/W025493/1). Gavin Abercrombie, Tanvi Dinkar and Verena Rieser were supported by the EPSRC project 'Gender Bias in Conversational AI' (EP/T023767/1). Tanvi Dinkar and Verena Rieser were supported by the EPSRC project 'AISEC: AI Secure and Explainable by Construction' (EP/T026952/1). Verena Rieser was also sup-

ported by a Leverhulme Trust Senior Research Fellowship (SRF/R1/201100). Amanda Cercas Curry was supported by the European Research Council (ERC) under the European Union's Horizon 2020 research and innovation program (grant agreement No. 949944, INTEGRATOR).

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
