# OpenReview forum: "Mirages. On Anthropomorphism in Dialogue Systems"
_EMNLP/2023/Conference — EMNLP 2023 Main_

### Official Review · Reviewer_dUKy · 2023-08-04

**Soundness:** 4

**Excitement:**

4: Strong: This paper deepens the understanding of some phenomenon or lowers the barriers to an existing research direction.

**Paper Topic And Main Contributions:**

This position paper examines the linguistic factors and psychological mechanisms that affect anthropomorphism in dialogue systems, the harms that may arise, and recommendations for dealing with these aspects. Throughout the paper, the authors also draw on existing efforts around anthropomorphism in dialogue and conversational systems. The main contribution is an organized collection of features, consequences, and recommendations surrounding anthropomorphism in dialogue systems.

**Questions For The Authors:**

A. Line 268 "People's expectation is that animate things---such as human beings---and inanimate ones---like machines---have very different functions and capabilities..." I wonder whether we can cleanly separate out these expectations nowadays? (Especially as dialogue and other conversational systems have already been integrated with our everyday tools, though it's true that new integrations are constantly forming)

B. "Responses to Direct Questions": from the paragraph title, I expected this section to go into more types of direct questions. Are there questions beyond "are you human" that warrant discussion?

C. Line 702 "We therefore recommend...forego embedding humanlike personality traits in dialogue systems" Are there exceptions? Contexts or scenarios in which the benefits of using humanlike personality traits may be greater than disadvantages (systems for loneliness, education, etc?) Even if the answer is no, it may be good to further contextualize.

**Reasons To Accept:**

Anthropomorphism in dialogue systems is an issue that is increasingly relevant, but there has not been much cohesive work in defining relevant linguistic and psychological features, consequences, and how/where we might begin to address these issues.
This paper ties in existing work and organizes a narrative that point the community to how we can take action to move away from anthropomorphic harms.

**Reasons To Reject:**

I don't really have a reason to reject this paper. Some of the paragraphs could be discussed in more detail, and it would be good to clarify the differences between definitions of "designers", "developers", "practitioners", etc (and thus perhaps the targeted audience of different recommendations) used throughout the paper, but those are more minor points.

**Reproducibility:**

N/A: Doesn't apply, since the paper does not include empirical results.

**Reviewer Confidence:**

3: Pretty sure, but there's a chance I missed something. Although I have a good feel for this area in general, I did not carefully check the paper's details, e.g., the math, experimental design, or novelty.

**Typos Grammar Style And Presentation Improvements:**

- Line 638: dollar sign instead of section?
- Line 686,687: missing spaces

---

> ### Author Rebuttal · Authors · 2023-08-28
>
> We thank the reviewer for their comments and questions.
>
> The final version of the paper will clarify/unify the terms "designers", "developers", "practitioners", which we currently use interchangeably to refer to anybody who may be in a position to make decisions on the features that we outline in the paper—from NLP researchers to product designers. We believe that the recommendations are worthy of consideration for anybody involved in the creation of dialogue systems designed to be interacted with by humans.
>
> Question A: this is an interesting question, worthy of future empirical work. In this paper we argue that, in order to avoid deception and the other potential harms that we outline, we should aim not to blur this line too much.
>
> Question B: this section does concern the many possible variations of the question “are you a human/machine?” as outlined by Gros et al. (2021), where we see the space of possible phrasings that allude to this as very broad. This contrasts with system output that may indirectly imply humanlike status, such as human activities, that we outline in subsequent subsections.
>
> Question C: while there may be situations (such as those the reviewer mentions) in which the benefits of anthropomorphism outweigh the risks, in writing the paper we didn’t encounter any evidence for this beyond the commercial reasons that we discuss in Section 2. In this position paper, we aim to convey a clear position that we should be careful when addressing social issues with technological solutions.
>
> Thanks for pointing out the typos, which we will correct for the final version of the paper.

---

### Official Review · Reviewer_n3SJ · 2023-08-05

**Soundness:** 4

**Excitement:**

4: Strong: This paper deepens the understanding of some phenomenon or lowers the barriers to an existing research direction.

**Paper Topic And Main Contributions:**

This position/survey paper addresses the anthropomorphism of NLP systems: it highlights the linguistic factors that influence people to anthropomorphize systems, discusses the negative consequences of this phenomenon, and makes some broad recommendations for how to address this issue in the future.

**Reasons To Accept:**

This is an excellently written paper that comprehensively unifies a lot of different research findings into an organized narrative. It will make a useful reference for any future work related to the topic. Some of the discussion points I find particularly insightful are:
- The cognitive factors that promote anthropomorphism, in particular effectance and sociality
- The fact that there is no one single feature of a system that will cause people to anthropomorphize it; instead, this emerges through multiple features
- The complicated role of some linguistic factors in anthropomorphism; for example, speech disfluencies can mitigate users’ over-confidence in a system but also make it appear more humanlike
- The importance of systems trying to reduce the tendency of users to anthropomorphize them, by avoiding language that presumes human qualities
- The recommendation to AI researchers to consider re-framing their research narratives to avoid references to building humanlike AI

**Reasons To Reject:**

I think this paper should be accepted for the reasons above. I do think the outlook expressed by the paper will be met with some opposing viewpoints, since it emphasizes the negative aspects of anthroporphism. A counter viewpoint is that anthropomorphism is not inherently harmful but simply needs to be controlled, and that some of recommendations given in the paper for avoiding it are unrealistic; for example, avoiding the use of first-person pronouns will be extremely difficult to achieve. I don’t see this as a weakness of the paper, but it may mean the authors may need to be prepared to further argue their position with even more detailed discussion (in either the presentation, and/or a follow-up publication, and/or some other venue).

**Reproducibility:**

N/A: Doesn't apply, since the paper does not include empirical results.

**Reviewer Confidence:**

4: Quite sure. I tried to check the important points carefully. It's unlikely, though conceivable, that I missed something that should affect my ratings.

---

> ### Author Rebuttal · Authors · 2023-08-28
>
> We thank the reviewer for their comments.
>
> We realise that some people are likely to dismiss the concerns raised in the paper, particularly due to the incentives to create anthropomorphic systems that we discuss in Section 2. However, the aim of our paper is to raise awareness of contributing factors to anthropomorphism that researchers and developers may not otherwise consider, and to suggest that systems are designed with these factors in mind. We do not seek to proscribe anthropomorphism, but rather to advocate for future research and development that is mindful of these issues. While it may be unrealistic to hope for developers of all future systems to ignore commercial pressures, we do not believe that it is impossible to develop NLP tools that are presented as tools rather than people. We realise that deployed systems may implement a pragmatic middle ground, but position papers such as this are meant to take a clear position and encourage discussion.

---

### Official Review · Reviewer_2Rub · 2023-08-06

**Soundness:** 4

**Excitement:**

4: Strong: This paper deepens the understanding of some phenomenon or lowers the barriers to an existing research direction.

**Paper Topic And Main Contributions:**

This is a positional paper arguing about the harms of Anthropomorphism in dialog systems.
1. The paper shared the Human, agent, and linguistic factors, and roles that contributed to Anthropomorphism.
2. The paper claimed the consequence of mis-trust & deception, giving machine genders.
3. This paper recommended 4 ways to reduce  anthropomorphism, including recognizing the tendency, using anthropomorphic tools, update research goal and avoid anthropomorphic system description.

The contributions include
1. Reemphasize the harms of anthropomorphism in dialogue system
2. Provided a detailed analysis of what are contributing to anthropomorphism
3. Gave practical recommendations of how to address the issue

**Reasons To Accept:**

1. Mainly based on the contribution above, with a focus of (2) and (3) in the contributions, as (1) isn't a new idea or arguments.
2. The paper cited a lot of references to show the point

**Reasons To Reject:**

(1) The harm of anthropomorphism in dialog system isn't a new thing, it isn't a new argument
(2) I would hope to see the analysis of anthropomorphism contributor can help guide the recommendation.
(3) In general, I don't feel I am convinced by this paper about the anthropomorphism's harm

**Reproducibility:**

N/A: Doesn't apply, since the paper does not include empirical results.

**Reviewer Confidence:**

3: Pretty sure, but there's a chance I missed something. Although I have a good feel for this area in general, I did not carefully check the paper's details, e.g., the math, experimental design, or novelty.

---

> ### Author Rebuttal · Authors · 2023-08-28
>
> We thank the reviewer for their comments.
>
> We do not claim to make a new argument that anthropomorphism in dialogue systems is harmful. Indeed, we cite several works that have made this claim before (e.g. West et al. 2019). Rather, we attempt to situate this problem in the specific context of the current NLP research and development with novel LLM-powered capabilities enabling anthropomorphic design to be more easily implemented. We believe that this topic is particularly timely due to the recent explosion in public awareness and wide-spread use of NLP systems. A novel contribution is our suggested taxonomy of how anthropomorphism is implemented and expressed within dialogue technology – detangling the phenomenon into Human, agent, and linguistic factors. This taxonomy aims to support more nuanced design decisions.
>
> It is unclear to us what causes the hesitation for the reviewer, but generally, our intention is to show contributing factors to anthropomorphism and draw attention to the implications of anthropomorphism, i.e., limited scepticism in the outputs of machine learning technologies.

---

### Meta-Review · Area_Chair_aSb6 · 2023-09-18

**Recommendation:** 4

**Metareview:**

This paper studies the harms of anthropomorphism in dialog systems. The authors studied cognitive factors that promote anthropomorphism, discussed relevant linguistic and psychological features causing this problem and made recommendations for reducing such problems. Reviewers agree that the paper has sufficient support for its claim and would be a great reference for future work in related topics.

---

### Decision · Program_Chairs · 2023-10-07

**Decision:**

Accept-Main

**Comment:**

This paper studies the harms of anthropomorphism in dialog systems. The authors studied cognitive factors that promote anthropomorphism, discussed relevant linguistic and psychological features causing this problem and made recommendations for reducing such problems. Reviewers agree that the paper has sufficient support for its claim and would be a great reference for future work in related topics.